# Two Pathway-Specific Transcriptional Regulators, PltR and PltZ, Coordinate Autoinduction of Pyoluteorin in *Pseudomonas protegens* Pf-5

**DOI:** 10.3390/microorganisms9071489

**Published:** 2021-07-13

**Authors:** Qing Yan, Mary Liu, Teresa Kidarsa, Colin P. Johnson, Joyce E. Loper

**Affiliations:** 1Department of Botany and Plant Pathology, Oregon State University, Corvallis, OR 97331, USA; Joyce.Loper@oregonstate.edu; 2Department of Plant Sciences and Plant Pathology, Montana State University, Bozeman, MT 59717, USA; mliu@montana.edu; 3Horticultural Crops Research Laboratory, US Department of Agriculture, Agricultural Research Service, Corvallis, OR 97330, USA; tkidarsa@gmail.com; 4Department of Biochemistry and Biophysics, Oregon State University, Corvallis, OR 97331, USA; Colin.Johnson@oregonstate.edu

**Keywords:** autoinduction, pyoluteorin, transporter, regulation, *Pseudomonas protegens*

## Abstract

Antibiotic biosynthesis by microorganisms is commonly regulated through autoinduction, which allows producers to quickly amplify the production of antibiotics in response to environmental cues. Antibiotic autoinduction generally involves one pathway-specific transcriptional regulator that perceives an antibiotic as a signal and then directly stimulates transcription of the antibiotic biosynthesis genes. Pyoluteorin is an autoregulated antibiotic produced by some *Pseudomonas* spp. including the soil bacterium *Pseudomonas protegens* Pf-5. In this study, we show that PltR, a known pathway-specific transcriptional activator of pyoluteorin biosynthesis genes, is necessary but not sufficient for pyoluteorin autoinduction in Pf-5. We found that pyoluteorin is perceived as an inducer by PltZ, a second pathway-specific transcriptional regulator that directly represses the expression of genes encoding a transporter in the pyoluteorin gene cluster. Mutation of *pltZ* abolished the autoinducing effect of pyoluteorin on the transcription of pyoluteorin biosynthesis genes. Overall, our results support an alternative mechanism of antibiotic autoinduction by which the two pathway-specific transcriptional regulators PltR and PltZ coordinate the autoinduction of pyoluteorin in Pf-5. Possible mechanisms by which PltR and PltZ mediate the autoinduction of pyoluteorin are discussed.

## 1. Introduction

Autoinduction, also called positive feedback regulation, is commonly used by bacteria to induce phenotypes required to respond promptly to a particular environmental condition [1,2,3]. The best-documented mechanism is the autoinduction of *N*-acyl-homoserine lactones, also known as quorum sensing signal molecules, which regulate cell-to-cell communication in many bacteria. In a classic model, *N*-acyl-homoserine lactone binds to its cognate response regulator and directly activates expression of *N*-acyl-homoserine lactone biosynthesis gene(s), which leads to an enhanced level of the signal molecule [4].

In addition to the quorum sensing signal molecules, production of many antibiotics is also controlled by autoinduction [5,6,7,8,9,10,11,12,13,14,15]. Through autoregulation, antibiotics can accumulate rapidly, allowing producers to respond promptly to competition with other microorganisms or predators [7]. Some antibiotics, such as 2,4-diacetylphloroglucinol [16] and jadomycins [14], bind to their cognate response regulators and directly induce expression of antibiotic biosynthesis genes, a mechanism similar to the autoinduction of quorum sensing signal molecules. For the vast majority of antibiotics, however, mechanisms of autoinduction are unknown.

This study focuses on the autoinduction of pyoluteorin, a broad-spectrum antibiotic produced by certain strains of *Pseudomonas* species including *P. aeruginosa* [17] and *P. protegens* (previously known as *P. fluorescens*) [18]. Autoinduction of pyoluteorin was first demonstrated in *P. protegens* Pf-5. Adding nanomolar concentrations of pyoluteorin to Pf-5′s cultures enhanced pyoluteorin production and transcription of pyoluteorin biosynthesis genes [9]. Later, pyoluteorin autoinduction was also observed in *P. protegens* CHA0 [19] and *P. aeruginosa* M18 [20]. Production of pyoluteorin requires a pathway-specific response regulator PltR (Figure 1) [21], which binds to the promoter region of pyoluteorin biosynthesis genes and activates their transcription [22]. PltR belongs to the LysR family regulators, which generally require activators for their activities [23]. Pyoluteorin was proposed to bind directly to PltR, thereby promoting PltR-mediated transcriptional activation of pyoluteorin biosynthesis genes and mediating autoinduction [22]. However, 2,4-dichlorobenzene-1,3,5-triol, a chlorinated phloroglucinol (PG-Cl_2_) with a chemical structure distinct from pyoluteorin (Figure 1A), is now known to be required for PltR-mediated activation of pyoluteorin production and its biosynthesis genes [24]. Mechanisms of pyoluteorin autoinduction remain unknown. Here we report that PltR is required but not sufficient for pyoluteorin autoinduction, and that PltZ, a second transcriptional regulator encoded in the *plt* gene cluster (Figure 1A), binds directly to pyoluteorin as an inducer and is also required for pyoluteorin autoinduction.

## 2. Materials and Methods

### 2.1. Strains and Cultural Conditions

The bacterial strains used in this study are listed in Table 1. *Pseudomonas protegens* Pf-5 and its mutants were cultured at 27 °C on King’s Medium B agar [25], Nutrient Agar (Becton, Dickinson and Company, Sparks, MD) supplemented with 1% glycerol (NAGly) or Nutrient Broth (Becton, Dickenson, and Company) supplemented with 1% glycerol (NBGly). Liquid cultures were grown with shaking at 200 r.p.m.

### 2.2. Construction of GFP Transcriptional Reporters and Expression Constructs

To measure the promoter activity of *pltZ*, a 235-bp DNA fragment containing *pltZ* promoter was amplified from the Pf-5 genome using oligonucleotide pair pltI-F4/pltI-R4 (Table 1), digested with *Eco*RI and *Sal*I, and ligated into pPROBE-NT to generate reporter construct pZ-*gfp* which contains the *pltZ*_promoter_:*gfp* transcriptional fusion. Similarly, a 574-bp DNA fragment containing *pltI* promoter was amplified using oligonucleotide pair pltI-F1/pltI-R1 (Table 1), digested with *Bam*HI and *Kpn*I, and ligated into pPROBE-NT to generate reporter construct p*I*-*gfp* which contains the *pltI*_promoter_: *gfp* transcriptional fusion.

The expression construct pME6010-*plt*Z was made by inserting a 697-bp *Xho*I-*Kpn*I PCR fragment containing the entire *pltZ* gene and a 21-bp 5′ untranslated region of *pltZ*, into pME6010 digested by the same restriction enzymes. This DNA fragment was amplified from the genomic DNA of Pf-5 using oligonucleotide pair pltZ-F2/pltZ-R2 (Table 1). The expression of *pltZ* gene was placed under the control of a constitutive kanamycin-resistance promoter (P*k*) carried by the vector pME6010 [33].

The expression construct pET28a-pltZ was made by inserting a 692-bp *Xho*I-*Nde*I PCR fragment containing *pltZ* gene into pET28a digested by the same restriction enzymes. This DNA fragment was amplified from the genomic DNA of Pf-5 using oligonucleotide pair pltZ 5′/pltZ 3′ (Table 1). Purified PltZ protein from the generated expression construct contains a 6xHis tag at the N-terminus. All these constructs were confirmed by Sanger sequencing analysis.

### 2.3. Construction of Pf-5 Mutants

The Δ*pltA*Δ*pltR* double mutant was made by deleting *pltR* from the chromosome of a Δ*pltA* mutant which was made previously [27]. To delete *pltR* gene from the chromosome of Pf-5, a *pltR* deletion construct P18Tc-Δ*pltR,* which was made previously [24], was transferred by conjugation from *E. coli* S17-1 to Pf-5 to delete 925-bp inside of the *pltR* gene in the chromosome of Pf-5. The deletion was confirmed by PCR analysis.

Similarly, the Δ*pltA*Δ*pltZ* double mutant was made by deleting *pltZ* from the chromosome of the Δ*pltA* mutant. To delete the *pltZ*, two DNA fragments flanking the *pltZ* gene were PCR amplified using the oligonucleotide pair pltZ UpF-Xba/pltZ UpR and pltZ DnR-Xba/pltZ DnF (Table 1). These two fragments were fused together by PCR and digested using *Xba*I to generate a 1003-bp DNA fragment containing *pltZ* with a 621-bp internal deletion. This DNA fragment was ligated to pEX18Tc to create construct p18Tc-ΔpltZ. This deletion construct was introduced into the Δ*pltA* mutant to make the Δ*pltA*Δ*pltZ* double mutant.

### 2.4. Assays for Monitoring GFP Activity of Reporter Constructs

The GFP-based transcriptional reporter assays were modified from a previous report [32]. Briefly, Pf-5 strains containing reporter constructs were cultured overnight in NBGly plus kanamycin (50 μg/mL) at 27 °C with shaking at 200 r.p.m. The cells were washed once into 1 mL fresh NBGly plus kanamycin with an optical density at 600 nm (OD_600_) of 1.0, and then used to inoculate 200 μL NBGly plus kanamycin to obtain a start OD_600_ of 0.01. Each strain was grown in three wells of a 96-well plate, which was incubated in a microplate shaker at 27 °C with shaking at approximately 400 r.p.m. Bacterial growth was monitored by measuring the OD_600_. The green fluorescence of bacteria was monitored by measuring emission at 535nm with an excitation at 485nm and corrected for background by subtracting fluorescence emitted by control strain which has an empty vector of the reporter construct.

### 2.5. Protein Purification and In Vitro Activity Assays

Protein overexpression and purification of PltZ was modified from previous method [24]. Briefly, the construct pET28a-pltZ was transformed into *E. coli* BL21 (DE3). Cells were cultured in LB broth with 50 μg/mL kanamycin at 37 °C to an OD_600_ of 0.5, then induced by 0.4 mM IPTG, and incubated overnight at 20 °C. The cells were lysed in Tris-NaCl buffer (20 mM Tris pH 7.0, 100 mM NaCl). The PltZ proteins with a N-terminal 6xHis tag were purified using a Ni-NTA Purification System (Invitrogen) and dialyzed in Tris-NaCl buffer. The protein concentrations were determined using a *DC* Protein Assay (Bio-Rad, Hercules, CA, USA).

Electrophoretic Mobility Shift Assay (EMSA) experiments were performed at 25 °C using a LightShift™ Chemiluminescent EMSA Kit (ThermoFisher). A 44-bp DNA probe named *pltIZ* (TATTTTCTAATTTAAATTCAAATTGAATTTTAATTAGGCCTTGG) and a 66-bp DNA probe named *pltRL* (TCGGGGCTGTTTTGCCTTTGCGGATATGCAAAGGCCTTTTGCAAAAACGGCTATTCACAAGTGCAT), that contain the putative promoter region of *pltI* and *pltL*, respectively, were synthesized and labeled with biotins using a Biotin 3′ End DNA Labeling Kit (ThermoFisher). Labeled DNA probes (1 nM) were mixed with different concentrations of PltZ protein and incubated at 25 °C for 20 min in a 20 μL reaction. If needed, unlabeled DNA probes (1 μM) were used to compete with the labeled DNA in binding PltZ. Various concentrations of pyoluteorin were added in the reaction to test if this compound can release free DNA probes from the protein-DNA complex. The reaction samples were loaded to a 5% native PAGE gel (Mini-PROTEAN^®^ TBE Gel, Bio-Rad) and electrophorized in 0.5× Tris Borate-EDTA (TBE) buffer at 80 V for one hour in a 4 °C cold room. The DNA samples were transferred onto a nylon membrane (Zeta-Probe^®^ Membrane, 9 × 12 cm, Bio-Rad) which then was exposed to a UV light treatment at 254nm for 10min to fix the nucleotides and the images were developed using X-ray films (CL-XPosure™ Film, ThermoFisher).

Interaction of PltZ with pyoluteorin was tested by two approaches. In the first approach, different concentrations (0–57.5 μM) of pyoluteorin were added into a PltZ protein solution (39 μM) which was prepared in a Tris-HCl buffer (20 mM Tris-HCL, pH 7.0). Absorbance of PltZ protein, pyoluteorin compound and their complex was measured in a 1-cm-path cuvette using a UV–vis spectrophotometer (Beckman DU730). The protein-compound solution which contains PltZ protein over saturated with pyoluteorin compound (57.5 μM) was loaded onto a Sephadex G-25 column (1 × 20 cm) preequilibrated and washed with 80 mL of the Tris-HCl buffer. The PltZ-pyoluteorin complex was eluted with 7 mL of the Tris-HCl buffer. The absorbance of purified PltZ-pyoluteorin complex was measured using the same method described above.

In the second approach, Isothermal titration calorimetry (ITC) experiments were performed at 25 °C using a MicroCal VP-ITC microcalorimeter (Malvern, US). Pyoluteorin was dissolved in methanol first and then diluted in the Tris-HCl buffer to a concentration of 100 μM. The titration assay included an initial 2 μL injection, followed by 44 (10 μL) injections of pyoluteorin into PltZ protein solution prepared above, accompanied by stirring at a constant rate. Protein samples and compound solutions were degassed by centrifugation at 14,000 r.p.m. for 20 min prior to the titration experiment. Data were processed using Origin 7.0 and fit to an independent binding model to calculate the thermodynamic parameters of interactions between PltZ and pyoluteorin.

## 3. Results

### 3.1. PltR Is Required But Not Sufficient for the Autoinduction of Pyoluteorin

To investigate the autoinduction of pyoluteorin, purified pyoluteorin was added into cultures of Pf-5 strains. A Δ*pltA* mutant was used to exclude the influence of internal pyoluteorin on the expression of pyoluteorin biosynthesis genes. The promoter activity of *pltL*, the first gene of the pyoluteorin biosynthesis gene cluster, was measured using the reporter construct p*L*-*gfp* which contains the *pltL* promoter fused with a promterless *gfp* (Figure 1B) [32]. Results show that amendment of pyoluteorin induced a significantly higher promoter activity of *pltL* in both the wild type Pf-5 and the Δ*pltA* mutant than in control cultures without pyoluteorin treatment (Figure 2A).

*pltR* encodes a transcriptional regulator known to activate the expression of pyoluteorin biosynthesis genes in Pf-5 (Figure 1) [21]. Consistent with the previous report, mutation of *pltR* in the Δ*pltA* mutant background markedly reduced *pltL* promoter activity (Figure 2A). Adding pyoluteorin into the Δ*pltA*Δ*pltR* mutant did not induce *pltL* promoter activity, indicating *pltR* is indispensable in pyoluteorin autoinduction (Figure 2A).

To test if *pltR* is sufficient for the autoinduction, we measured *pltL* promoter activity in *P. fluorescens* SBW25 using a reporter construct p*RL*-*gfp,* which contains the *pltR* gene and the *pltL* promoter fused with a promoterless *gfp* (Figure 1C) [24]. Strain *P. fluorescens* SBW25 does not contain pyoluteorin biosynthesis genes [34] but, if *pltR* is introduced, strain SBW25 has the genetic capability to activate *pltL* transcription as proven by induced *pltL* promoter activity with PG-Cl_2_ treatment (Figure 2B). Results show that pyoluteorin could not induce *pltL* promoter activity in SBW25/*pRL-gfp* (Figure 2B), indicating that *pltR* is not sufficient for pyoluteorin autoinduction. To test the possibility that PltR is sufficient for autoinduction in the presence of both pyoluteorin and PG-Cl_2_, both compounds were added to the cultures of SBW25/p*RL*-*gfp*. Results show that pyoluteorin did not enhance *pltL* expression in the presence of PG-Cl_2_ (Figure 2B).

Collectively, these results indicate that PltR is indispensable for the transcription of pyoluteorin biosynthesis genes, but an additional regulatory factor(s) is(are) needed for autoinduction of pyoluteorin.

### 3.2. PltZ Is an Autorepressor and Directly Represses Expression of Linked Transporter Genes

In addition to *pltR*, the *plt* gene cluster encodes another transcriptional regulator PltZ (Figure 1B). Homologs of *pltZ* genes repress the expression of linked transporter genes in *P. aeruginosa* strains M18 [35] and ATCC 27853 [36]. The substrate of the encoded transporter (termed PltIJKNOP in *P. protegens* Pf-5) has not been demonstrated, and the function of PltZ in *P. protegens* has not been characterized. In this work, *pltZ* was deleted in the Δ*pltA* mutant of Pf-5, which generated a Δ*pltA*Δ*pltZ* double mutant. Two reporter constructs, p*Z*-*gfp* and p*I*-*gfp*, containing the promoter of *pltZ* and *pltI* respectively fused with a promoterless *gfp*, were made to test if PltZ regulates the expression of itself or the linked transporter genes *pltIJKNOP* (Figure 1). Relative to Δ*pltA* mutant, Δ*pltA*Δ*pltZ* mutant showed a significantly higher *pltZ* promoter activity (Figure 3A), indicating that PltZ negatively regulates its own transcription. We further tested *pltZ* promoter activity in strain *P. fluorescens* SBW25 carrying a functional *pltZ* on the expression plasmid pME6010-*pltZ*. Results show that *pltZ* markedly decreased its own transcription in SBW25 (Figure 3B), suggesting that PltZ is an autorepressor that directly regulates the transcription of itself.

Using the reporter construct p*I*-*gfp,* which contains *gfp*-fused promoter of *pltI*, the first gene of the *pltIJKNOP* transcriptional unit [37], we show that *pltI* promoter activity was significantly higher in the Δ*pltA*Δ*pltZ* mutant than in the Δ*pltA* mutant (Figure 4A). These results indicate that PltZ represses the transcription of *pltIJKNOP* transporter genes in Pf-5. Further, *pltZ* significantly reduced *pltI* promoter activity in strain SBW25 (Figure 4B), indicating that PltZ directly represses transcription of *pltIJKNOP* transporter genes.

We then purified PltZ protein and tested if PltZ binds to the *pltI* promoter in vitro. Results of gel shifting assays show a clear shift of the labeled *pltIZ* DNA probe which contains *pltI* promoter (Figure 5A,B), but not the *pltRL* DNA probe that contains *pltL* promoter (Figure 5C,D), in the presence of PltZ protein, thus confirming the direct regulation of the *pltIJKNOP* transcriptional unit by PltZ. The PltZ-*pltIZ*_probe_ protein-DNA complex can be disintegrated by pyoluteorin at micromolar concentrations, that are within the physiological range of Pf-5 [24], as revealed by the released free DNA probe in the presence of pyoluteorin. This result shows that pyoluteorin reduces the binding affinity of PltZ to the promoter region of the *pltIJKNOP* transcriptional unit.

Collectively, our results show that PltZ is an autorepressor and directly represses the transcription of *pltIJKNOP* transporter genes in strain Pf-5.

### 3.3. Pyoluteorin Interacts with PltZ and Relieves PltZ-Mediated Transcriptional Repression of pltIJKNOP Transporter Genes

PltZ belongs to the TetR family of regulators that usually interact with small molecules or protein ligands, which then change their regulatory activities [38]. We have shown that pyoluteorin can release the DNA probe that contains the *pltI* promoter region from the PltZ-*pltIZ*_probe_ complex (Figure 5AB). To test if pyoluteorin serves as an inducer to relieve PltZ-mediated repression of the *pltIJKNOP* transcriptional unit, pyoluteorin was added into cultures of strain SBW25 containing the reporter construct p*I*-*gfp* and the expression plasmid pME6010-*pltZ*. Our result show that pyoluteorin significantly increased *pltI* promoter activity (Figure 6A), suggesting that pyoluteorin is likely an inducer of PltZ.

We then tested the interaction of pyoluteorin and PltZ using two independent approaches. In the first spectrophotometer assay, adding pyoluteorin to a PltZ protein solution resulted in a new absorbance peak at around 375 nm (Figure 6B), likely caused by the PltZ-pyoluteorin complex. Continued addition of pyoluteorin generated a second new absorbance peak at around 315 nm, which is identical to the free pyoluteorin compound. Elution of the protein-compound solution through G25 column maintained the absorbance peak at around 375 nm, but markedly reduced absorbance at around 315 nm (Figure 6B, inner figure), likely due to the removed free pyoluteorin compound that was retained on the column. The binding of pyoluteorin to PltZ was further evaluated in an isothermal titration calorimetry (ITC) assay. Adding pyoluteorin compound into PltZ protein solution markedly changed the protein conformation of PltZ at a molar ratio around 0.5 (Figure 6C). Data fitting revealed that pyoluteorin binds to PltZ with a binding affinity of around 13.2 nM.

Together, these results demonstrate that pyoluteorin binds PltZ as an inducer, relieving the PltZ-mediated transcriptional repression of *pltIJKNOP* transporter genes.

### 3.4. PltZ Is Involved in the Autoinduction of Pyoluteorin

Based on our results that pyoluteorin interacts with PltZ to derepress transporter genes in the pyoluteorin gene cluster, in addition to the known coordinate regulation of the biosynthesis and transport genes [37], we hypothesized that PltZ is involved in the autoinduction of pyoluteorin. To test this hypothesis, the p*L*-*gfp* reporter construct was moved into the Δ*pltA*Δ*pltZ* mutant, and *pltL* promoter activity of the resultant reporter strain was measured with or without pyoluteorin treatment. Mutation of *pltZ* markedly increased the promoter activity of *pltL* (Figure 7), which is consistent with the previous report that *pltZ* negatively regulates pyoluteorin production of *P. aeruginosa* M18 [39]. Importantly, *pltL* promoter activity could not be further induced by pyoluteorin amendment in the Δ*pltA*Δ*pltZ* mutant, indicating that PltZ is necessary for pyoluteorin autoinduction.

## 4. Discussion

Understanding the regulation of antibiotic production is of great interest not only in basic microbiology but also in clinical, industrial, and agricultural microbiology. Autoinduction is a widespread regulatory mechanism of antibiotic production, allowing producing microorganisms to quickly accumulate antibiotics in response to changing environmental conditions [7]. Here, using *P. protegens* Pf-5 as a model, we report an alternative mechanism of autoinduction by which two pathway-specific transcriptional regulators PltR and PltZ are both required for the autoinduction of the antibiotic pyoluteorin.

Both PltR and PltZ are required for pyoluteorin autoinduction but they play different regulatory roles. The requirement of PltR is consistent with previous reports that PltR protein directly binds to *pltL* promoter and activates transcription of pyoluteorin biosynthesis genes in *P. aeruginosa* M18 [22], and that mutation of *pltR* abolishes pyoluteorin production in *P. protegens* Pf-5 [21]. Together, these data show that PltR plays an indispensable role in pyoluteorin autoinduction because it is required for expression of pyoluteorin biosynthesis genes. Although PltZ is also required for pyoluteorin autoinduction, it is not necessary for the basal levels of expression of pyoluteorin biosynthesis genes or pyoluteorin production. Instead, mutation of *pltZ* enhanced *pltL* transcription in Pf-5 (Figure 7). Similarly, mutation of *pltZ* increased pyoluteorin yield and transcription of pyoluteorin biosynthesis genes in M18 [39]. These results suggest that the pathway-specific regulation of pyoluteorin biosynthesis gene expression involves at least two processes: a basal level of regulation (transcriptional activation) mediated by PltR and a higher level of regulation (autoinduction) mediated by PltZ. This mechanism differs from the known mechanisms of antibiotic autoinduction, which involves a single pathway-specific transcriptional regulator that perceives an antibiotic (or a biosynthetic intermediates) as a signal and directly stimulates transcription of the antibiotic biosynthesis genes [14,16].

The mechanisms by which PltZ mediates pyoluteorin autoinduction were not investigated in this study. However, we proved that pyoluteorin alters PltZ-mediated direct regulation of *pltIJKNOP* transcription with the following evidences: (1) *pltZ* repressed *pltI* promoter activity heterogeneously in strain SBW25 (Figure 4B); (2) pyoluteorin relieved PltZ-mediated repression of *pltI* promoter activity in SBW25 (Figure 6A); (3) PltZ bound a *pltI*_promoter_-containing DNA probe in vitro; and (4) pyoluteorin released a *pltI*_promoter_-containing DNA probe bound by PltZ (Figure 5).

The direct interaction between pyoluteorin and PltZ and the direct regulation of *pltIJKNOP* transporter gene expression by PltZ, together with the result that mutation of *pltZ* abolished autoinduction of pyoluteorin (Figure 7), prompt us to hypothesize that the PltIJKNOP transporter is involved in pyoluteorin autoinduction. Although the substrate(s) of this transporter remain(s) unidentified, it is known to be required for a normal production level of pyoluteorin: mutation of the transporter encoding genes decreased pyoluteorin yield in strains Pf-5 and M18 [37,40]. Bioinformatic analysis indicates that the PltIJKNOP transporter system likely serves as an exporter in pumping out intracellular metabolites [37,40]. One possible mechanism of pyoluteorin autoinduction is that the PltIJKNOP transporter, activated by pyoluteorin, exports and reduces intracellular level of metabolite(s) that is(are) repressive to transcription of pyoluteorin biosynthesis genes. Repressive metabolite(s) can be generated before pyoluteorin biosynthesis. For example, phloroglucinol (PG), the precursor of PG-Cl_2_ [24], is known to inhibit transcription of pyoluteorin biosynthesis genes at micromolar concentrations via an uncharacterized mechanism [41]. Repressive metabolite(s) may also be generated during pyoluteorin biosynthesis. Intermediates of many antibiotic biosynthesis pathways have been found to function as signaling molecules in self-protection and/or production of resultant metabolites [42,43,44]. Future efforts are needed to investigate if the PltIJKNOP transporter is involved in the PltZ-mediated autoinduction of pyoluteorin, and if PG and/or other potential repressive metabolites of pyoluteorin biosynthesis can be exported by the PltIJKNOP transporter and their roles in the autoinduction of pyoluteorin.

Another possible mechanism by which PltZ controls pyoluteorin autoinduction is that PltZ may directly bind to *pltL* promoter and repress the expression of pyoluteorin biosynthesis genes. Pyoluteorin may relieve this repression, resulting in an enhanced transcription of pyoluteorin biosynthesis genes and antibiotics production. A 16-bp semi-palindromic PltZ-binding DNA motif (TNNAATTNNAATTNNA) identified in *P. aeruginosa* ATCC 27853 [36] is also present in the *pltI* promoter region of Pf-5 (Figure 5A). However, this PltZ-binding DNA motif was not found in the *pltL* promoter region (Figure 5C), and no interaction was observed between PltZ and a DNA fragment containing the intergenic region between *pltR* and *pltL* of Pf-5 (Figure 5D) or *P. aeruginosa* ATCC 27853 [36], suggesting that PltZ is unlikely to regulate *pltL* directly.

In summary, we report that pyoluteorin serves as an inducer that binds to PltZ and relieves PltZ-mediated repression of *pltIJKNOP* transcription. Results of this study also show that *pltZ*, together with the other pathway-specific transcriptional regulator *pltR*, is essential for autoinduction of pyoluteorin in Pf-5.

## Figures and Tables

**Figure 1 microorganisms-09-01489-f001:**
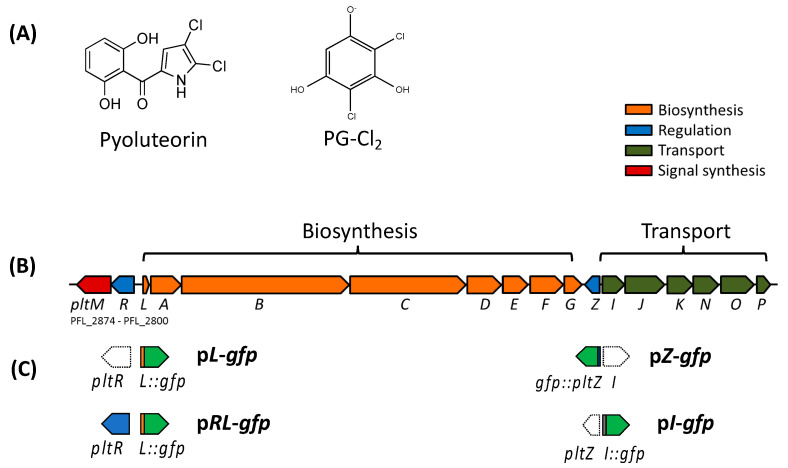
Chemical structures of pyoluteorin and PG-Cl_2_ (2,4-dichlorobenzene-1,3,5-triol) (**A**), pyoluteorin biosynthesis gene (*plt*) cluster of Pf-5 (**B**), and *promoter:: gfp* fusions of the transcriptional reporter constructs used in this work (**C**). Figures are modified from previous publication [24]. The *plt* gene cluster contains two transcriptional regulatory genes *pltR* and *pltZ*. PG-Cl_2_, an activator of PltR, is synthesized by PltM with phloroglucinol (PG) as a substrate. Open arrows in (**C**) indicate the genes are not included in the related constructs.

**Figure 2 microorganisms-09-01489-f002:**
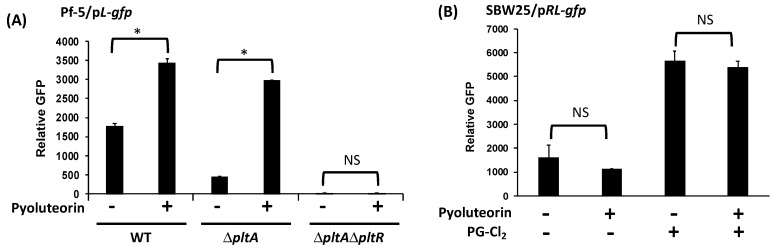
Autoinduction of pyoluteorin biosynthesis genes requires *pltR*. (**A**) *pltL* promoter activity was measured using the GFP reporter construct p*L*-*gfp* in wild type (WT) Pf-5 and its mutants. (**B**) *pltL* promoter activity was measured using the GFP reporter construct p*RL*-*gfp* in strain SBW25. The promoter activity was measured as relative GFP (fluorescence of GFP divided by OD_600_) recorded at 24 h (**A**) and 15 h (**B**) post inoculation (hpi). Pyoluteorin and PG-Cl_2_ were added to the cultures at a concentration of 1 μM and 10 nM, respectively. The inducing effect of PG-Cl_2_ on *pltL* promoter activity has been reported previously [24] and is confirmed in this study. * indicates treatments are significantly different (*p* < 0.05), as determined by the Student *t*-test. NS: not significant. Data are means and standard deviations of at least three biological replicates from a representative experiment repeated three times with similar results.

**Figure 3 microorganisms-09-01489-f003:**
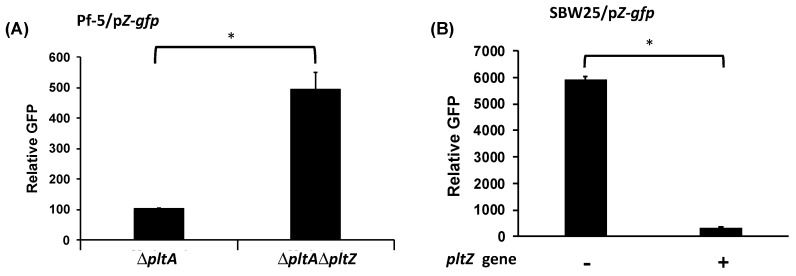
Autorepression of *pltZ*. Promoter activity of *pltZ* was measured using the GFP reporter construct p*Z*-*gfp* in Pf-5 mutants (**A**), or in strain SBW25 with or without the presence of the expression construct pME6010-*pltZ* (**B**). Promoter activity was measured as relative GFP recorded at 24 hpi. * indicates treatments are significantly different (*p* < 0.05), as determined by the Student’s *t*-test. Data are means and standard deviations of at least three biological replicates from a representative experiment repeated three times with similar results.

**Figure 4 microorganisms-09-01489-f004:**
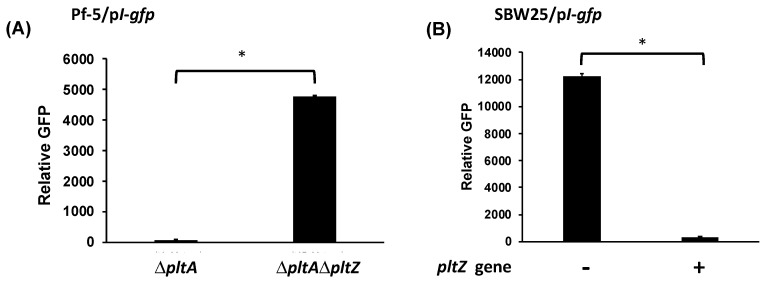
Repression of *pltI* promoter activity by *pltZ*. Promoter activity of *pltI* was measured using the GFP reporter construct p*I*-*gfp* in Pf-5 mutants (**A**), or in strain SBW25 with or without the presence of expression construct pME6010-*pltZ* (**B**). Promoter activity was measured as relative GFP recorded at 24 hpi. * indicates treatments are significantly different (*p* < 0.05), as determined by the Student’s *t*-test. Data are means and standard deviations of at least three biological replicates from a representative experiment repeated three times with similar results.

**Figure 5 microorganisms-09-01489-f005:**
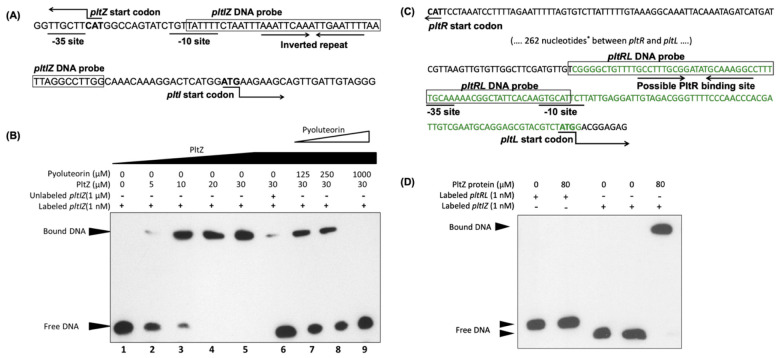
PltZ binds to the *pltI-pltZ* intergenic region. (**A**) Intergenic region between *pltI* and *pltZ* in the chromosome of Pf-5. The predicted -10 and -35 sites of *pltI* promoter and an inverted repeat are shown. (**B**) EMSA binding assay of interactions between PltZ protein, the *pltIZ* DNA probe (shown in (**A**)), and the pyoluteorin compound. The interactions were performed in 20 μL reactions at 25 °C. Lane 1 serves as a control which contains only labeled *pltIZ* DNA probe; lanes 2 to 5 contain labeled *pltIZ* probe and different concentrations of PltZ protein; lane 6 contains unlabeled *pltIZ* DNA fragment to compete for PltZ protein; lanes 7 to 9 contain different concentrations of pyoluteorin to test if this compound releases the *pltIZ* probe from its binding complex with PltZ. (**C**) Intergenic region between *pltR* and *pltL* in the chromosome of Pf-5. The predicted −10 and −35 sites of *pltL* promoter and a possible PltR binding site similar to the *lys* box identified in *P. aeruginosa* M18 [22] are shown. *: 262 nucleotides between the indicated *pltR* and *pltL* intergenic region are not shown. The DNA fragment labeled with green color was fused with a promoterless *gfp* to monitor the promoter activity of *pltL* in the reporter construct p*L*-*gfp* that was made previously [32] and used in this study (Figure 1C). (**D**) EMSA binding assay of interactions between PltZ protein and the *pltRL* DNA probe (shown in (**C**)). The *pltIZ* DNA probe used in (**B**) served as a positive control. The experiments were repeated at least two times.

**Figure 6 microorganisms-09-01489-f006:**
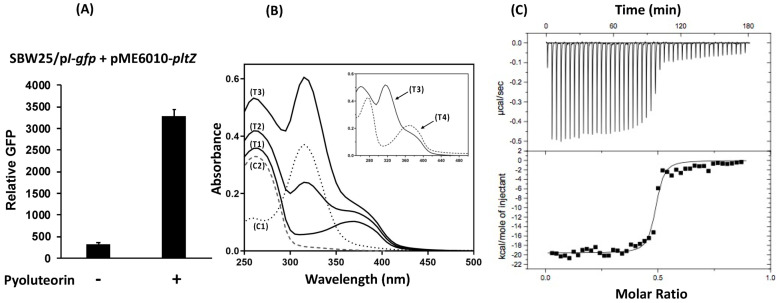
Pyoluteorin relieves PltZ-mediated repression on *pltI* promoter activity (**A**) and binds to PltZ protein in vitro (**B**,**C**). (**A**) Promoter activity of *pltI* was measured in strain SBW25 containing the GFP reporter construct p*I*-*gfp* and the expression construct pME6010-*pltZ* with or without the addition of pyoluteorin (1 μM). The promoter activity was measured as relative GFP recorded at 24 hpi. Data are means and standard deviations of at least three biological replicates from a representative experiment repeated three times with similar results. (**B**) Pyoluteorin was added into PltZ protein solution in Tris-HCl buffer at different concentrations. The absorbance profile of the mixtures was monitored. C1: control protein solution that has free purified PltZ protein. C2: control compound solution that has free purified pyoluteorin compound. T1 to T3: mixtures of PltZ protein (39 μM) and pyoluteorin compound at 7.5 μM (T1), 32 μM (T2), 57.5 μM (T3). T4 (inner figure): filtrates of T3 eluted from a Sephadex G-25 column. (**C**) Binding isotherm for the pyoluteorin-PltZ interaction. The integrated heat was plotted against the molar ratio between the ligand and the protein. Data fitting revealed a binding affinity of about 13.2 nM. The experiments of (**B**,**C**) were repeated two times.

**Figure 7 microorganisms-09-01489-f007:**
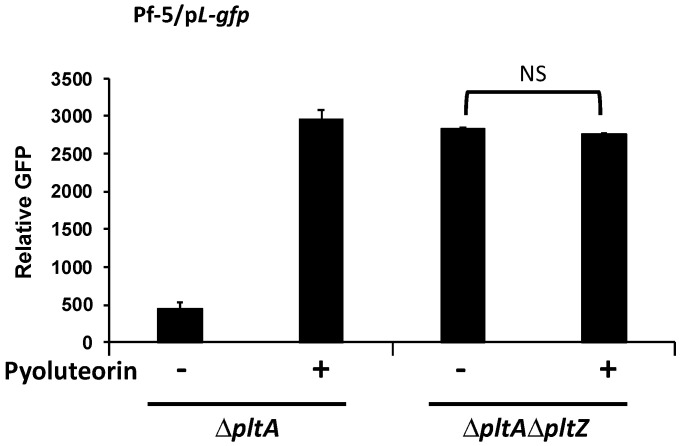
Autoinduction of pyoluteorin biosynthesis genes requires *pltZ*. *pltL* promoter activity was measured using the GFP reporter construct p*L*-*gfp* in Pf-5 mutants. The promoter activity was measured as relative GFP recorded at 24 hpi. Pyoluteorin was added to the cultures at a concentration of 1 μM. NS: not significant (*p* > 0.05), as determined by the Student’s *t*-test. Data are means and standard deviations of at least three biological replicates from a representative experiment repeated three times with similar results.

**Table 1 microorganisms-09-01489-t001:** Bacterial strains, plasmid, and primers used in this study.

Strains, Plasmids, or Primers	Genotype, Relevant Characteristics or Sequences	Reference or Source
Strains ^#^		
*P. protegens*		
LK099	Wild-type strain Pf-5.	[26]
JL4805	Pf-5 derivative strain contains an in-frame deletion of *pltA* in the chromosome.	[27]
LK530	Pf-5 derivative strain contains an in-frame deletion of *pltA* and *pltR* in the chromosome. This mutant was generated by deleting *pltR* in JL4805.	This study
LK419	Pf-5 derivative strain contains an in-frame deletion of *pltA* and *pltZ* in the chromosome. This mutant was generated by deleting *pltZ* in JL4805.	This study
*P. fluorescens* SBW25	SBW25 is a member of the *P. fluorescens* group that does not contain *plt* gene cluster.	[28]
*E. coli*		
S17-1	*recA pro hsdR*^−^M^+^ RP4 2-Tc::Mu-Km::Tn7 Sm^r^ Tp^r^	[29]
BL21 (DE3)	*omp*T *hsd*S_B_ (r_B_^−^m_B_^−^) *gal dcm*	NEB
Plasmids		
pEX18Tc	Gene replacement vector with MCS from pUC18, *sacB*^+^ Tc^r^	[30]
pEX18Tc-Δ*pltR*	pEX18Tc containing *pltR* with a 925-bp in-frame deletion.	[24]
pEX18Tc-Δ*pltZ*	pEX18Tc containing *pltZ* with a 621-bp in-frame deletion.	This study
pPROBE-NT	pBBR1, containing a promoterless *gfp*, Km^r^	[31]
p*L*-*gfp*	Called ppltL-*gfp* previously. Contains the intergenic region between *pltR* and *pltL* including *pltL* promoter fused with a promoterless *gfp*.	[32]
p*RL*-*gfp*	Contains *pltR* and the intergenic region between *pltR* and *pltL* including *pltL* promoter fused with a promoterless *gfp*.	[24]
p*I*-*gfp*	Contains the intergenic region between *pltI* and *pltZ* including *pltI* promoter fused with a promoterless *gfp*.	This study
p*Z*-*gfp*	Contains the intergenic region between *pltI* and *pltZ* including *pltZ* promoter fused with a promoterless *gfp*.	This study
pME6010	pACYC177-pVS1 shuttle vector, Tc^r^	[33]
PME6010-*pltZ*	pME6010 with a 697-bp *Xho*I-*Kpn*I PCR fragment amplified from the genomic DNA of Pf-5, containing a constitutively expressed *pltZ* gene.	This study
Primers *	oligonucleotide sequences (5′to 3′)	
pltZ-F2	ACTCGAGAAAATAACAGATACTGGCCATG	
pltZ-R2	ATAGGTACCGACTATTGGGCAATGGC	
pltI-F1	ATAGGATCCAGCAGACGGAATTTGGGC	
pltI-R1	TATGGTACCCCTACAATCAACTGCTTCTTC	
pltI-F4	TGGCGAATGTCGACGTTGC	
pltI-R4	TATGAATTCCTCGTGGTCTGAGCGG	
pltZ 5′primer	GTCATACATATGAAGCAACCCCCCGCTC	
pltZ 3′primer	ATGATCTCGAGCCGACTATTGGGCAATG	
pltZ UpF-Xba	CTCCTCTCTAGATCAAAACGTGCCTTGGACTG	
pltZ UpR	ACTATTGGGCAATGATGTTCCTCGTGGTCTGAG	
pltZ DnF	ACGAGGAACATCATTGCCCAATAGTCGGCTCA	
pltZ DnR-Xba	CACACCTCTAGAACATCCCCTGCTGTTTCTTC	

^#^ abbreviations of antibiotics and their concentrations used in this work are: Tc, tetracycline (10 μg/mL for *E. coli*, 200 μg/mL for Pf-5); Km, kanamycin (50 μg/mL). * underlines show DNA restriction enzyme sites that were used for cloning. All primers were designed in this study.

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
