# Peer review of "Two Pathway-Specific Transcriptional Regulators, PltR and PltZ, Coordinate Autoinduction of Pyoluteorin in Pseudomonas protegens Pf-5"

_microorganisms, 2021, doi:10.3390/microorganisms9071489_

Round 1
Reviewer 1 Report
Very interesting work with impressive results. It will greatly help other researchers who would like to work on regulation in antibiotic production. Only minor changes are required regarding the use of the English language. I think it is ready for publication.
Author Response
Thank you for your review! We corrected a few typos and words to improve the use of English language in the revised manuscript.
Reviewer 2 Report
In this manuscript, the authors have presented interesting results, namely, an alternative mechanism of antibiotic autoinduction by which the two pathway-specific transcriptional regulators PltR and PltZ in pyoluteorin production in Pseudomanas protegens Pf-5.
Not only the involvement of two pathway-specific transcription factors in antibiotic autoinduction, but the clear demonstration that one of the pathway-specific transcription factors, PltZ, is involved in the regulation of transporter genes is valuable because it provides a new perspective on the understanding of antibiotic autoinduction.
Therefore, I consider this manuscript to be suitable for publication in Microorganisms.
However, there remain a few concerns, which are as follows:
- Table 1: Isn’t it necessary to show Pseudomonas fluorescens (protegens?) SBW25 strains in Table 1?
- Line 198, etc.: If Pseudomonas fluorescens has been renamed to P. protegens, shouldn't the species names in the text be unified?
- Fig. 5(B): In EMSA, is the pyoluteorin concentration at which PltZ and DNA are sufficiently dissociated possible physiologically?
- Lines 300-301: Authors describes that “pyoluteorin interacts with PltZ to repress transporter genes”. Is this correct?
- Lines 350-351: Authors describes that “mutation of pltZ abolished autoinduction of pyoluteorin (Fig. 3)”. Does Figure 3 show this?
- Lines 375-378: Authors describes that “However, this PltZ-binding DNA motif was not found in the pltL promoter region, and no interaction was observed between PltZ and a DNA fragment containing the pltR and pltL intergenic region of P. aeruginosa ATCC 27853, suggesting that PltZ is unlikely to regulate pltL directly”. Although this discussion is reasonable, since the proposed PltZ-binding DNA motif was identified in P. aeruginosa ATCC 27853, which is different species from the species used in this study, authors should perform experiments to confirm whether PltZ binds to the promoter region of pltL of P. protegens Pf-5 that was used in this study.
Author Response
In this manuscript, the authors have presented interesting results, namely, an alternative mechanism of antibiotic autoinduction by which the two pathway-specific transcriptional regulators PltR and PltZ in pyoluteorin production in Pseudomanas protegens Pf-5.
Not only the involvement of two pathway-specific transcription factors in antibiotic autoinduction, but the clear demonstration that one of the pathway-specific transcription factors, PltZ, is involved in the regulation of transporter genes is valuable because it provides a new perspective on the understanding of antibiotic autoinduction.
Therefore, I consider this manuscript to be suitable for publication in Microorganisms.
Response: Thank you for your review!
However, there remain a few concerns, which are as follows:
- Table 1: Isn’t it necessary to show Pseudomonas fluorescens(protegens?) SBW25 strains in Table 1?
Response: Yes. Information of P. fluorescens SBW25 has been added in Table 1 of the revised manuscript.
- Line 198, etc.: If Pseudomonas fluorescenshas been renamed to P. protegens, shouldn't the species names in the text be unified?
Response: One lineage of the Pseudomonas fluorescens group, including strain Pf-5 used in this study, was renamed to a new species P. protegens due to the distinct characteristics of the group (Ramette et al., 2011). However, many strains of Pseudomonas fluorescens including strain SBW25 retain the species name (P. fluorescens), as used in this manuscript.
Ramette, A., Frapoilli, M., Fischer-Le Saux, M., Gruffax, C., Meyer, J.-M., Défago, G., et al. (2011) Pseudomonas protegens sp. nov., widespread plant-protecting bacteria producing the biocontrol compounds 2,4-diacetylphloroglucinol and pyoluteorin. System Appl Microbiol 34: 180–188.
- 5(B): In EMSA, is the pyoluteorin concentration at which PltZ and DNA are sufficiently dissociated possible physiologically?
Response: In our EMSA, micromolar concentrations of pyoluteorin are sufficient to dissociate the PltZ-pltIZprobe protein-DNA complex. Our previous study (Yan et al., 2017) indicated that Pf-5 can produce pyoluteorin at micromolar concentrations that are similar with the pyoluteorin concentrations used in the EMSA of this study (Fig. 5B). This information has been added to the revised manuscript (lines 295-297).
Yan, Q., Philmus, B., Chang, J.H., and Loper, J.E. (2017) Novel mechanism of metabolic co-regulation coordinates the biosynthesis of secondary metabolites in Pseudomonas protegens. eLife 6: e22835.
- Lines 300-301: Authors describes that “pyoluteorin interacts with PltZ to repress transporter genes”. Is this correct?
Response: This sentence has been corrected to “pyoluteorin interacts with PltZ to derepress transporter genes” in the revised manuscript (lines 370-371).
- Lines 350-351: Authors describes that “mutation of pltZabolished autoinduction of pyoluteorin (Fig. 3)”. Does Figure 3 show this?
Response: The correct figure is Figure 7. This sentence has been corrected in the revised manuscript (line 429).
- Lines 375-378: Authors describes that “However, this PltZ-binding DNA motif was not found in the pltLpromoter region, and no interaction was observed between PltZ and a DNA fragment containing the pltR and pltL intergenic region of P. aeruginosa ATCC 27853, suggesting that PltZ is unlikely to regulate pltL directly”. Although this discussion is reasonable, since the proposed PltZ-binding DNA motif was identified in P. aeruginosa ATCC 27853, which is different species from the species used in this study, authors should perform experiments to confirm whether PltZ binds to the promoter region of pltL of P. protegens Pf-5 that was used in this study.
Response: The interaction between PltZ and a pltRL DNA probe containing the promoter region of pltL of P. protegens Pf-5 was tested. The result showed that PltZ does not bind to the promoter region of pltL of P. protegens Pf-5. The result has been added to Figure 5 (C and D), lines 281-285 and 453-457 in the revised manuscript.